# Iowa Newborn Screening Program Experience with Hemoglobinopathy Screening over the Last Two Decades and Its Increasing Global Relevance

**DOI:** 10.3390/ijns10010021

**Published:** 2024-03-08

**Authors:** Ryan Jilek, Jennifer Marcy, Carol Johnson, Georgianne Younger, Amy Calhoun, Moon Ley Tung

**Affiliations:** 1Carver College of Medicine, University of Iowa, Iowa City, IA 52242, USA; 2Stead Family Department of Pediatrics, Division of Medical Genetics and Genomics, Iowa City, IA 52242, USA; 3Iowa Newborn Screening Program, Iowa City, IA 52242, USA

**Keywords:** newborn screening program, hemoglobinopathy, sickle cell disease (SCD), non-sickling disease, thalassemia

## Abstract

Hemoglobinopathies are the commonest monogenic disorder worldwide, with approximately seven percent of the world population being carriers of hemoglobinopathies. The healthcare utilization impact of thalassemia has resulted in global public health initiatives to screen for hemoglobinopathies, especially sickle cell disease (SCD). The Iowa Newborn Screening Program (INSP) has been in place for more than 50 years with a primary focus on detecting SCD. Recent changes in migration patterns have led to a global distribution of hemoglobinopathies in the western world, which has translated to an increase in the diagnosis of SCD and the incidental detection of non-sickling hemoglobinopathies within the INSP. This study documents the birth prevalence of hemoglobinopathies diagnosed in newborns through the INSP and highlights the need for newborn screening programs to evolve to meet the healthcare needs of underserved, minority populations.

## 1. Introduction

The Iowa Newborn Screening Program (INSP) has been in place since 1966 to screen all babies born in Iowa for congenital diseases, which, if left undetected and untreated, can have serious implications, such as disability or death. Newborn screening tests include a hearing screen, pulse oximetry for critical congenital cardiac defects, and a dried blood spot (DBS) screen for genetic or congenital disorders. Currently, the DBS screen includes almost 50 congenital disorders such as organic acidemias, fatty acid oxidation disorders, amino acidemias, urea cycle disorders, endocrinopathies, cystic fibrosis, classic galactosemia, biotinidase deficiency, severe combined immunodeficiency, spinal muscular atrophy, selected peroxisomal and lysosomal disorders, and hemoglobinopathies.

Hemoglobinopathy screening was implemented in Iowa in 1989, and its focus is to identify sickle cell disease (SCD), which includes sickle cell anemia (HbSS), hemoglobin SC disease (HbSC), and sickle-beta thalassemia (HbSβ). These sickling hemoglobinopathies are a major health problem, and there are approximately more than 100,000 patients with SCD in the United States [1,2]. Recent global estimates indicate an annual incidence of 300,000 babies born with the sickle hemoglobin (HbS) allele, and it is expected to increase to 400,000 by 2050, with the majority of cases in sub-Saharan Africa, the Middle East, and India [3]. Locally, the National Institutes of Health (NIH) recommended universal screening for SCD in 1987 based on the benefit provided by penicillin prophylaxis against life-threatening pneumococcal infection in infants with SCD [4]. The successes of universal newborn screening for SCD in the United States are manifold and include a decrease in mortality by 50% in children aged one to four years, an increase in the overall life expectancy, and the identification of at-risk couples who would benefit from prenatal genetic counseling [5]. Currently, all 50 states and the District of Columbia perform newborn screening for SCD as recommended by the American College of Medical Genetics (ACMG) and endorsed by the March of Dimes, given the positive public health impacts [6,7]. According to the yearbook of immigration statistics in 2022 (https://www.dhs.gov/immigration-statistics/yearbook/2-22#test, accessed on 18 September 2023), total immigration to the United States has increased, especially from places that have high rates of thalassemia, and as a result, newborn screening programs are identifying non-sickling hemoglobinopathies that are more prevalent in Asian populations, such as alpha [8] and beta thalassemia [9] in addition to SCD. This trend is reflected in other developed countries experiencing similar demographic shifts with significant impacts on newborn screening policies [10]. Certain subgroups of these non-sickling hemoglobinopathies lead to non-transfusion dependent thalassemia (NTDT), such as Hemoglobin H disease (HbH) or reduced expression of the beta globin genes, resulting in an imbalance of the alpha to beta globin chains, such as Hemoglobin E/beta thalassemia (HbE/β thalassemia) [11]. Although NTDT does not require regular blood transfusions for their daily function, it is increasingly recognized that this entity leads to significant morbidities such as growth retardation, pulmonary hypertension, gallstones, leg ulcers, splenomegaly, liver disease, thromboembolic events, extramedullary hematopoiesis, and iron overload state [12,13]. A recent publication demonstrated that NTDT receiving transfusion therapies have an independently higher risk of red cell alloimmunization rates compared to transfusion-dependent thalassemia (TDT) [14].

Despite the progress made in universal newborn screening programs in the United States, there are publications documenting a wide variability in state-based recommendations for screening methodologies, follow-up of non-sickling hemoglobinopathies [15], and reporting of alpha thalassemia silent carrier and trait status identified through the newborn screening process [8,16]. Seven of the fifty states do not report presumed alpha thalassemia silent carrier or trait-identified results on the newborn screen to providers or family. Twelve states report these results with recommendations for routine care, and the remaining thirty-one states include management strategies beyond routine care, although there is significant variation in the type and timing of laboratory studies based on the individual patient’s demographics and clinical findings.

In this study, we discuss the birth prevalence of abnormal newborn screens for hemoglobinopathy detected through INSP within a 20-year period and the molecular characterization of these patients. We found an increasing trend of SCD since 2015, which may be attributable to migrants from Africa. Although the birth prevalence of non-sickling hemoglobinopathies was low at 0.02%, molecular testing in 39% of these cases revealed a significant proportion of HbH disease (18%). These results provide data to better understand the public health impact and clinical outcomes of hemoglobinopathies in the state of Iowa and function as a stepping stone in the formulation of a standardized algorithm for screening methodologies, diagnosis, and short- and long-term follow-up recommendations for hemoglobinopathies detected through the INSP.

## 2. Materials and Methods

### 2.1. Patients and Study Design

This retrospective, observational study evaluates all newborn screening samples performed by the State Hygienic Laboratory (SHL) for the state of Iowa from 1 January 2000 to 31 December 2020. This study was approved by the local institutional review board (IRB ID #: 202102526). All newborns had a heel prick and dried blood spots (DBS) collected on filter paper (PerkinElmer Inc, Waltham, MA, USA.) with 5 pre-printed 3.2 mm internal diameter circle between 24 and 48 h of age, according to the Clinical and Laboratory Standards Institute (CLSI) guidelines for blood collection on filter paper for newborn screening programs. Information that is collected on the Iowa newborn screening cards includes the collection date and time, initial versus repeat screen, collector details, infant details (medical record number, first and last name, birth time and date, gender, zip code, gestational age at birth, feeding method, current weight in grams, transfusion history, intensive care unit admission, presence or absence of meconium ileus), guardian’s details (first and last name, date of birth, gender, phone number), ordering healthcare provider’s details (name, phone number, national provider identifier number), and primary care provider’s details (first and last name, phone number). For infants who have been transfused prior to newborn screening, a repeat collection is recommended at around 56 days post-transfusion. For preterm newborns who were born prior to 37 weeks of gestational age, a repeat collection is recommended. These cards are couriered to the SHL and processed daily. Quality checks are performed on the DBS cards, and the data are entered into the local integrated database called the Iowa Newborn Screening Information System (INSIS).

### 2.2. Isoelectric Focusing (IEF)

The first-tier method of screening for hemoglobinopathy is performed via isoelectric focusing (IEF) (RESOLVE^TM^, Revvity, Waltham MA, USA), which separates various hemoglobin in a sample. One 3.2 mm DBS is treated with Hb elution solution to elute hemoglobin and inhibit the formation of methemoglobin F. The eluted sample is applied to the pre-cast agarose gel containing mixtures of low-molecular-weight amphoteric molecules (pH 6–8) with varying isoelectric points in a sample application template. An electric current is applied to the gel, and both the amphoteric and hemoglobin molecules migrate across the gel based on their individual isoelectric points. Migration ceases when the charges on the variants are zero. The electric field counteracts diffusion, and the hemoglobin variant forms a discrete thin band. Different hemoglobin have different isoelectric points (pI), resulting in various bands observed across the gel. The isoelectric points of the sample can be compared to standard values to determine the type of hemoglobin each band represents. The bands are stained, and the amount of each type of hemoglobin can be found using densitometry, which determines the density of each band, quantifying the results. By using densitometry, it can be determined if the sample has abnormal amounts of hemoglobin. IEF is more accurate than other electrophoresis methods, as the bands produced are sharper and can detect more types of hemoglobin.

### 2.3. High-Performance Liquid Chromatography (HPLC)

High-performance liquid chromatography (HPLC) is performed as a second-tier method when the results of the IEF show that hemoglobin A (HbA) is absent, an abnormal band is present, or Hb Barts is greater than or equivalent to 10.5%. HPLC can identify common hemoglobin variants, including, but not limited to, HbA, HbA2, HbF, Hb Barts, HbS, HbE, and HbC [17]. Eluates of the DBS specimen are processed using the VARIANT nbs Automatic Sampler (VARIANT nbs Newborn Screening System, Bio-Rad, Hercules, CA, USA). Specimens are sequentially injected into the analysis stream and separated by the analytical cartridge. Different types of hemoglobin elute at various times due to differences in charge. Spectroscopy is used to determine retention times, and a chromatogram is obtained. The observed retention times are then compared to standard retention times to determine which type of hemoglobin each peak represents. The areas under the curve of the chromatogram are calculated to determine the percent of each type of hemoglobin present in the sample, and using this, different forms of hemoglobin can be detected.

### 2.4. Short-Term Follow-Up Process

The short-term follow-up team at INSP will inform the birthing hospital or the primary care physician of newborns with positive newborn screening results for SCD and non-sickling hemoglobinopathy with recommendations for a referral to a pediatric hematology–oncology unit for further evaluation and management. SCD cases are closed once it has been confirmed that the newborn has been seen by a pediatric hematologist and started on penicillin prophylaxis. The current short-term follow-up standard operating procedure for non-sickling hemoglobinopathies varies based on the variant hemoglobin and the percentage of Hb Barts detected (Figure 1 and Figure 2). The INSP uses an adult hemoglobin (HbA) level of less than 3% as an action value to report the screening as present for fetal hemoglobin (Hb F) only. Confirmed cases are reported to the birthing hospitals via the HL-7 reporting system and entered into a national repository, NewSTEPs, by the short-term follow-up team. Confirmatory molecular testing is not performed by the INSP and is sent at the discretion of the evaluating pediatric hematologist.

### 2.5. Data Collection

All abnormal newborn screens for SCD and non-sickling hemoglobinopathy performed at the Iowa SHL between 1 January 2000 and 31 December 2020 were collected retrospectively using the INSP database. Although the Iowa SHL performs newborn screening in the states of Iowa, North Dakota, South Dakota, and Alaska, only data from the state of Iowa were analyzed in this study. The following information was recorded for all samples: sample ID, gender, date of birth, ethnicity, IEF and HPLC results, and final INSP diagnosis. Molecular genotype was recorded if it was performed, and the results were reported to the INSP team.

## 3. Results

A total of 808,596 newborns were screened during the study period from 1 January 2000 to 31 December 2020. Patient demographics are summarized in Table 1. Our results show a birth prevalence of 151 positive newborn screens for SCD (Figure 3) and 179 positive newborn screens for non-sickling disease (Figure 4) detected using the Iowa NBS program during this period. There is an increasing trend of SCD since 2015, which may be due to a recent increase in migrants from Asia and Africa.

Amongst the 179 cases of non-sickling disease, there were 41 false-positive cases. These were mainly premature infants with a positive screen on initial testing and a normal result on a repeat sample. Two cases were pending at the time of data analysis, and there were six cases with missing data. Thirteen cases were lost to follow-up, and two cases were recorded as against medical advice. At the closing of their cases, seven were deceased, and all seven cases showed only fetal hemoglobin (Hb F) on the IEF and HPLC screens. Molecular testing was performed for 38.5% of all the abnormal newborn screens for non-sickling disease and showed a variety of alpha and beta globin variants, such as beta-thalassemia major (Homozygous β^0^β^0^), homozygous hemoglobin E (Hb EE), deletional, and non-deletional Hb H disease. The genotyping effort was mainly supported by collaborative research with the Children’s Hospital of Oakland Research Institute (CHORI) in California. Overall, there were 4 cases of beta-thalassemia major (2.2%), 65 cases of beta-thalassemia with structural variants (36.3%), 32 cases of Hb H disease (17.9%), and 7 cases of digenic thalassemia detected in this study (Table 2). A summary of these details is presented in a flowchart (Figure 5). Amongst the 65 cases of beta thalassemia and its structural variants, only 28 (43%) of these cases were confirmed molecularly (Table 2). Genotyping was performed for two-thirds of the 32 Hb H cases, and the majority of these cases were due to a deletional Hb H disease secondary to the compound heterozygote state for the Southeast Asian (SEA) deletion type of α^0^-thalassemia with the 3.7 kb deletion of the alpha globin gene (α^3.7^). Four cases were inferred based on prior laboratory findings in first-degree relatives.

## 4. Discussion

Hemoglobinopathies are a group of disorders attributable to genetic variants that result in either the production of abnormal structural hemoglobin(s) or thalassemia, which refers to the absence of normal alpha or beta globin chain production [18]. Structural variants such as hemoglobin S (Hb S), hemoglobin C (Hb C), hemoglobin D (Hb D), or hemoglobin E (Hb E) lead to a qualitative defect, whereas thalassemia results in a quantitative defect due to unbalanced globin synthesis [19,20]. A point mutation in the adult beta globin gene (*HBB*) (c.20A>T, p.Glu7Val) results in the production of an abnormal hemoglobin (Hb S), which polymerizes when deoxygenated and leads to the classical clinical phenotype seen in sickle cell disease (SCD) [21]. Beta thalassemia can be caused by a variety of molecular variants, ranging from point mutations to small deletions limited to the adult beta globin gene (*HBB*) (OMIM *141900) to large deletions involving the whole beta globin gene cluster. Conventional hemoglobin nomenclature denotes beta thalassemia alleles as β^0^ when no beta globin is produced and β^+^ when less than normal beta globin chains are produced [19]. On the other hand, alpha thalassemia is mainly attributed to deletions in the alpha hemoglobin locus 1 (*HBA*1) (OMIM *141800) or alpha hemoglobin locus 2 (*HBA*2) (OMIM *141850) [20]. Normal adult hemoglobin (Hb A) contains two alpha and two beta globin chains (α2/β2), which are encoded by the *HBA*1/*HBA*2 and *HBB* genes, respectively. Normally, an individual has a total of four alpha globin genes, with two located on each copy of chromosome 16 [20]. Individuals with three normal alpha genes are silent carriers (α^+^-thalassemia trait or −α/αα), whereas those with two normal alpha genes are classified as alpha thalassemia minor. Those with two alpha gene deletions in the trans form are noted as homozygous α^+^-thalassemia trait (−α/−α), and those with gene deletions in the cis form are noted as α^0^-thalassemia trait (αα/−−) [22]. Hb H disease occurs when there is a deletion of three alpha genes due to a compound heterozygote state (α^+^/α^0^ or −−/−α), leading to a reduction in the alpha globin gene expression to less than 30% of normal values [20] with a concurrent increase in beta globin chain production. Unstable β4 tetramers (Hb H) are formed due to this increase, which precipitates within erythrocytes [23]. Non-sickling hemoglobinopathies can be caused by alpha, beta, or a combination of alpha and beta thalassemia. A deletion of all four alpha genes results in the homozygous form of α^0^-thalassemia (−−/−−), which leads to the formation of gamma chains (γ4) tetramers, also known as Hb Bart’s [24]. Hb Bart’s is the most severe form of alpha thalassemia and leads to hydrops fetalis, resulting in either intrauterine death or early post-natal mortality.

This retrospective study identified 179 abnormal newborn screens for non-sickling hemoglobinopathies and 151 positive newborn screens for SCD within the state of Iowa over a 20-year period. This translates into a birth prevalence of 2.2/10,000 for non-sickling hemoglobinopathies and 1.9/10,000 for SCD, which would be relatively high for a midwestern region within the United States. Although newborn screening for SCD has been in place since 1987, the overall birth prevalence of SCD and non-sickling hemoglobinopathies within the United States is not known [13,25]. A recent prospective study evaluating newborn screening in Berlin, Germany, described a high SCD birth prevalence of 2.4/10,000 [26] with seven molecularly confirmed cases (four cases of homozygous HbSS, two compound heterozygotes for HbS/C, and one compound heterozygote for hemoglobin S and hereditary persistence of fetal hemoglobin (HbS/HPFH)). Information on ethnicity was available for only six of these cases, with five cases from West Africa and one from Central Africa.

Based on the 2020 State Data Census (https://www.iowadatacenter.org/index.php/data-by-source/decennial-census/race-and-hispanic-or-latino-origin-1990-2020, accessed on 26 October 2022), approximately 17.3% of Iowa’s population of 3,190,369 are non-White, which has increased from 4.1% since 1990. Thus, this birth prevalence rate is consistent with the migration patterns observed within the United States. Although information on ethnicity is not routinely collected on the DBS cards, the short-term follow-up team recorded this information when available during the infant’s first evaluation with the pediatric hematologist at our center. One unique feature of the INSP is the availability of a genetic counselor who provides genetic counseling at one of the two pediatric hematology clinics in Iowa during the newborn’s first hematology evaluation. This allowed us to record the ethnicity of 12% of the positive newborn screens for non-sickling hemoglobinopathies, and as expected, most of these cases were of Southeast Asian ancestry (6%).

The molecular information in these non-sickling hemoglobinopathy cases is important, as certain variant hemoglobin (such as Hb E, Hb C, HbD, or Hb D-Iran) can result in NTDT and even thalassemia major when co-inherited with beta thalassemia [13,15]. Deletional Hb H disease refers to the deletion removing both alpha globin genes on one chromosome 16 such as the SEA type of α^0^-thalassemia plus a deletion removing only a single alpha globin gene on the other chromosome 16 such as the 3.7 kb deletion (α^3.7^) or the 4.2 kb deletion (α^4.2^) [27]. Non-deletional Hb H disease, on the other hand, refers to the deletion removing both alpha globin genes on one chromosome 16 plus an α^+^-thalassemia due to a point mutation or small insertion or deletion involving either the *HBA1* or *HBA2* alpha globin gene on the other chromosome. There is wide phenotypic variability in Hb H disease, and the severity of anemia depends on the underlying molecular subtype. Non-deletional Hb H disease, such as Hb H Constant Spring (α^CS^), has a more severe clinical phenotype and presents with lower hemoglobin levels compared to deletional Hb H disease. Newborns with Hb H Constant Spring have a higher percentage of Hb Bart’s and develop multi-system complications within the first decade of life [13,15,22,27].

The NIH recommended universal screening for SCD in 1987, and implementation of this universal screening was adopted by all 50 states, the District of Columbia, and many United States territories by 2006 [18]. The implementation of universal newborn screening for SCD is variable across states with respect to screening, reporting, and referral of non-sickling hemoglobinopathies, as these are considered secondary screening targets [25]. Most states would refer newborns with a screening test that shows only the fetal hemoglobin (Hb F) pattern or elevated Hb Bart’s for confirmatory molecular testing [28]. Screening methodologies employed by each state also vary, but most newborn screening laboratories use HPLC and/or IEF as a two-tiered screening approach, in line with the current recommendations by the Agency for Health Care Policy and Research (AHCPR) [25]. Only three states in the United States employ molecular-based analysis for second-tier testing, and they include New York, Texas, and Washington [25]. The INSP uses a two-tiered approach, using IEF as the first screening method, followed by HPLC as the second complementary method.

A nationwide survey of state-based newborn screening programs within the United States in August 2018 documented significant variation in the screening methodology and result reporting and highlighted the need for standardization of newborn screening programs to ensure health equity [9]. In this report, 86% of the newborn screening programs use specific cut-off levels to define beta thalassemia, but there was significant variation in these levels, with some programs using a cutoff of Hb F only (or less than 1% of Hb A) and others utilizing a threshold of equal or less than 1% to less than 3% of Hb A. Recent data from the United Kingdom showed that a Hb A value of less than 1.5% has a good sensitivity, specificity, and positive predictive value of 98.5%, 99.9%, and 91.7% for beta thalassemia detection in newborn screening [29], with only one false-negative case presenting with a Hb A level of 1.7%. This baby received clinical care and was confirmed to have beta thalassemia major, following a known increased risk based on parental testing results. In this study, all false-positive cases were attributed to prematurity of 32 weeks gestational age or less. A similar nationwide survey of newborn screening programs in 2016 documented significant variability in the screening methods and result reporting for alpha thalassemia as well [8]. The INSP reports possible Hb H disease when the Hb Barts levels are greater than 25% and a possible alpha thalassemia carrier state if the Hb Bart’s level is between 10.5 and 24.9% (Figure 2). Based on the 2016 survey, reasons indicated for not reporting elevated Hb Bart’s levels were attributed to the lack of a HPLC setup, the lack of technology to confirm or quantify Hb Bart’s level, the absence of a commercially available Hb Bart’s standard, and the fact that alpha thalassemia is not on the Recommended Uniform Screening Panel (RUSP) [8]. The development of a national hemoglobinopathy screening protocol with standardized cut-off levels to identify thalassemia can be important in helping quickly identify hemoglobinopathies, ensuring quick and accurate treatment.

To improve patient outcomes, it is imperative to have long-term follow-up in place if a newborn screen is positive for hemoglobinopathy. This can include establishing quality measures for newborn screening programs, such as accurate data collection, robust disease-specific registries, and the initiation of a long-term follow-up program. For example, the Cystic Fibrosis Foundation (CFF) has played an important role in long-term treatment by using quality measures to collect accurate patient data for cystic fibrosis [30]. The Cystic Fibrosis Foundation Patient Registry (CFFPR) has over 48,000 patients and is highly accurate, with over 90% of hospitalizations and clinical visits recorded [30]. The Registry and Surveillance System for Hemoglobinopathies (RuSH) was formed by the CDC in 2004 to collect data on hemoglobinopathies, including thalassemia, in hopes of improving treatment. However, data were collected for only seven states, and by 2008, only two states continued to collect data until 2015 [31]. During this period, over 130,000 hemoglobinopathies were reported to the database for data collection [31]. In 2015, the CDC funded a new program titled Sickle Cell Data Collection (SCDC) to collect data on patients with sickle cell disease. As of 2021, 11 states are participating in the data collection [32]. Although the SCDC is not collecting data on other hemoglobinopathies, it is promising that more programs are being established to help improve long-term care for hemoglobinopathies.

An example of long-term care is a program that the American College of Medical Genetics and Genomics (ACMG) has recently launched called the Long-Term Follow-Up Cares and Check Initiative (LTFU Cares and Check) (https://longtermfollowupnbs.org accessed on 15 June 2023). The focus of the program is to enable newborns who are diagnosed with a genetic disease through newborn screening programs to achieve the best clinical outcome through the development of a systematic and comprehensive long-term model system. This program will focus their initial efforts on targeting Spinal Muscular Atrophy (SMA), but it is with great anticipation that we will see hemoglobinopathy as one of the target diseases of this initiative to improve the health and clinical outcomes of patients with hemoglobinopathies, regardless of their race, ethnicity, or geographical location. Registries with robust quality measures can help identify variables such as socioeconomic factors, health disparities, and optimal treatment algorithms for a disease to improve outcomes in patients with hemoglobinopathies.

### Novel Health Initiatives in Thalassemia

Since the treatment of thalassemia requires complex protocols, accurate and timely information can help both patients and clinicians further understand the success of treatment protocols and activities of daily living. Mobile health applications may be a potential novel health initiative that can be implemented in the care of patients with thalassemia. Over 85% of Americans own a smartphone, a 50% increase since 2011 [33]. Mobile health apps (mHealth) are currently being used for many chronic diseases. In the management of chronic obstructive pulmonary disease (COPD), one of the leading causes of death in the United States, it was found that the app, Wellkins mHealth, improved patient outcomes and COPD management [34]. Another example where mHealth is shown to improve patient outcome is in cardiovascular disease, the leading cause of death in the United States [35]. Various mHealth apps can increase physical activity in users with cardiovascular disease [36]. Launched in 2017, chemoWave is an app that allows patients with cancer to track medication usage, daily living activities, and register their symptoms. The individual datum collected from the user provides patients and clinicians with insights into a patient’s daily symptoms and reactions to treatment regimens and allows for individualized suggestions around diet and activity. Oncologists have even used chemoWave to help find the best time for a patient’s next chemotherapy cycle [37]. Although there are many apps for various chronic diseases that enable patient interaction, there are very few apps made for thalassemia. In a study that researched 88 various health apps for hemoglobinopathies, only 1 was designed for thalassemia [38]. An app called ThaliMe, launched in 2018, is focused on helping connect users with thalassemia. Developers hope to raise awareness for the disease and improve patient outcomes with data collection; however, ThaliMe is limited to specific countries and is currently not available for download in the United States. Treatment of thalassemia requires various treatment modalities, a network of caregivers, and cooperation across multiple medical specialties. Leveraging technology that enables clear insight into patients impacted by hemoglobinopathies needs to be readily available, easily understood by patients, and include standardized history collection to enable patients and clinicians to better understand pharmacologic and non-pharmacologic impacts in their long-term care.

The main limitation in this study is the retrospective nature, which results in missing data as well as the lack of a long-term follow-up program that systematically collects clinical, molecular, and laboratory data and outcomes from pediatric hematologists. The lack of national guidelines that recommend molecular confirmation in all newborns with abnormal newborn screens for hemoglobinopathies also makes the collection and interpretation of clinically relevant public health data challenging.

This study represents a good starting point for the initiation of prospective studies that systematically collect long-term follow-up data from newborn screening programs, especially in infants with hemoglobinopathy, to accurately document the exact birth prevalence of SCD and non-sickling hemoglobinopathies within the United States in the face of a rapidly changing population demographic driven by global migration patterns.

## Figures and Tables

**Figure 1 IJNS-10-00021-f001:**
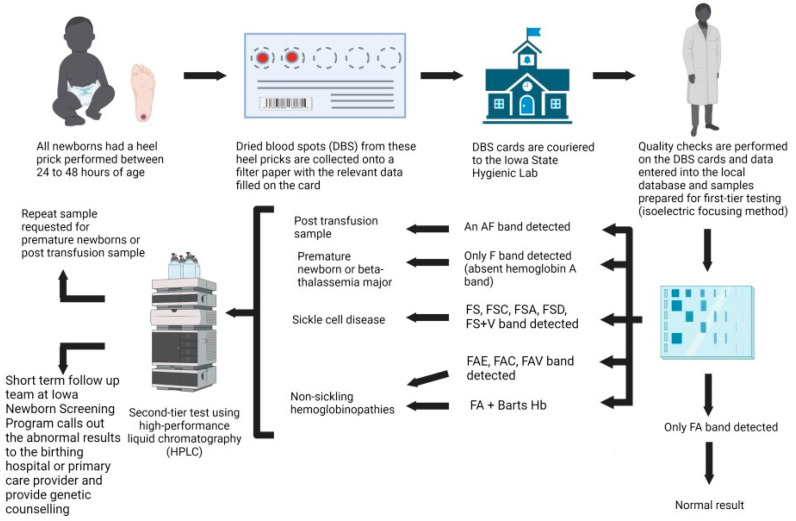
Standard operating procedures for the Iowa Newborn Screening Program (INSP) for hemoglobinopathies. Figure created with BioRender.com.

**Figure 2 IJNS-10-00021-f002:**
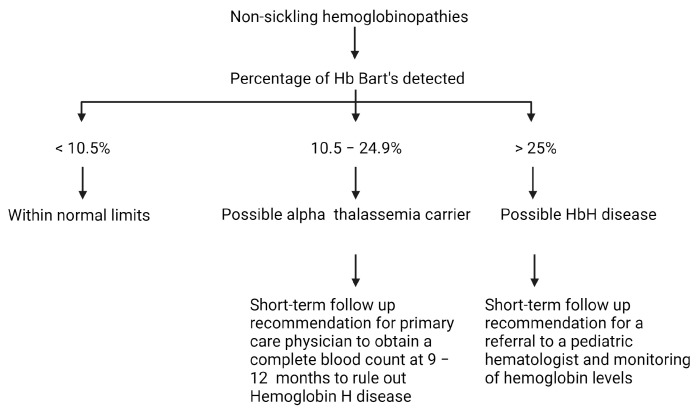
Iowa Newborn Screening Program (INSP) short-term follow-up protocol for non-sickling hemoglobinopathies. Figure created with BioRender.com.

**Figure 3 IJNS-10-00021-f003:**
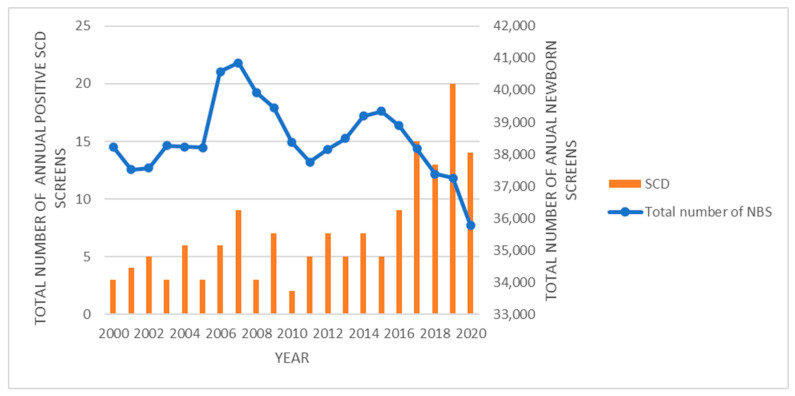
Total number of annual positive newborn screens for sickle cell disease (SCD) in comparison to the total number of annual newborn screens for the state of Iowa through the Iowa Newborn Screening Program from 2000 to 2020.

**Figure 4 IJNS-10-00021-f004:**
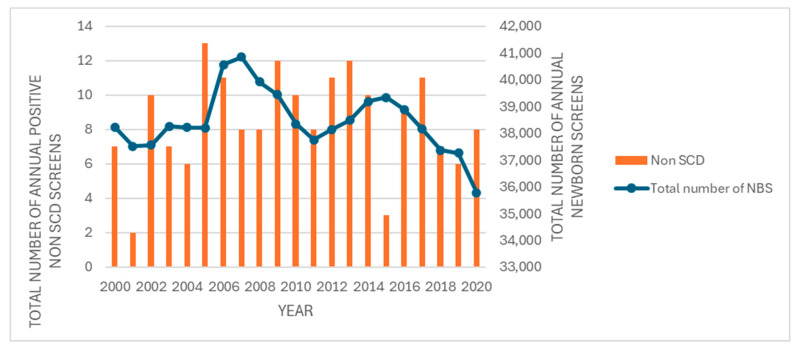
Total number of annual positive newborn screens for non-sickling hemoglobinopathy (Non-SCD) in comparison to the total number of annual newborn screens for the state of Iowa through the Iowa Newborn Screening Program from 2000 to 2020.

**Figure 5 IJNS-10-00021-f005:**
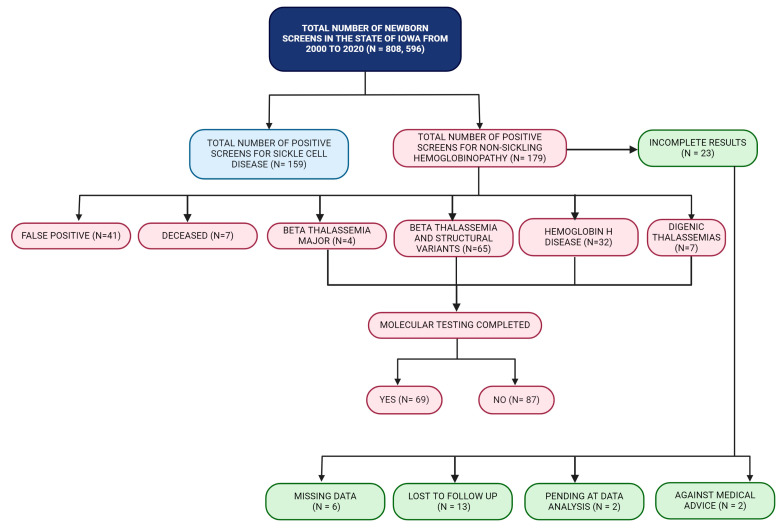
Summary flowchart of the positive newborn screens for non-sickling hemoglobinopathies. Figure created with BioRender.com.

**Table 1 IJNS-10-00021-t001:** Patient demographics for positive newborn screens for non-sickling hemoglobinopathy.

Demographics	N (%)
Total number of positive newborn screens for non-sickling hemoglobinopathies identified	179 (100)
Gender	
Male	101 (56)
Female	78 (44)
Ethnicity	
Missing data	158 (88)
Chinese	1 (0.5)
Burmese	5 (3)
Laotian	4 (2)
Cambodian	1 (0.5)
Togolese	1 (0.5)
Saudi Arabian	1 (0.5)
Turkish	1 (0.5)
Caucasian	2 (1)
African American/ Caucasian	2 (1)
Asian/Laotian	1 (0.5)
Asian/ Mexican	1 (0.5)
Nepalese/ Bhutanese	1 (0.5)
Confirmatory molecular testing performed	
Yes	69 (39)
No	110 (61)

**Table 2 IJNS-10-00021-t002:** Genotype results for positive newborn screens for non-sickling hemoglobinopathy.

MOLECULAR GENOTYPE	N
**Beta thalassemia major (Homozygous β^0^ β^0^)** HBB: c.92+1G>T and 3.5 kb deletion HBB: Homozygous c.-137C>A HBB: Homozygous c.315+1G>A Parental testing: Both with beta thalassemia trait	1 1 1 1
**Beta thalassemia and structural variants** Homozygous HbE: HBB: c.79G>A HBB: negative HbE/β^0^ thalassemia HBB: c.79G>A and c.126_129delCTTT Parental testing: Homozygous HbEE and beta thalassemia trait Sibling with HBB: c.79G>A and c.126_129delCTTTT Heterozygous D trait (HbAD): HBB: Heterozygous HbD Homozygous Hb C: HBB: c.19G>A Heterozygous G-Coushatta trait: HBB c.68A>C HbD-Iran/β^0^ thalassemia: HBB: c.67G>C and c.92+1G>A Compound heterozygous HbS/ Hb Athens-GA: HBB: c.20A>T and c.122G>A Heterozygous C trait (HbAC): HBB: negative Homozygous Hb D: HBB: c.364G>C	14 2 1 1 1 1 2 2 1 1 1 1
**HbH disease** Deletional HbH α^3.7^/SEA α^4.2^/ alpha2 gene deletion α^3.7^/ heterozygous 2 gene deletion Parental testing: Both have an alpha globin gene deletion Non-deletional HbH HBA2: c.*92A>G and –^MED-II^ SEA/α^CS^ (HBA2: c.427T>C, p.X143Gln) SEA/HBA1: c.389T>C SEA/HBA1: c.62_63insT	13 1 1 1 1 2 2 1
Digenic thalassemias Homozygous HbE/α^CS^α^CS^ Homozygous HbE/α^3.7^ Homozygous HbE/α^3.7^ SEA Heterozygous HbE/α^3.7^ SEA Beta thalassemia/ Hb Westmead trait: HBB: E and IVSII-836A>G (VUS); HBA2: heterozygous c.369C>G	1 1 1 1 1
Incomplete molecular data	10
**Total**	**69**

## Data Availability

The original contributions presented in the study are included in the article, further inquiries can be directed to the corresponding author.

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
