# Peer review of "Iowa Newborn Screening Program Experience with Hemoglobinopathy Screening over the Last Two Decades and Its Increasing Global Relevance"

_2409-515X, 2024, doi:10.3390/ijns10010021_

Round 1

Reviewer 1 Report

Comments and Suggestions for Authors

This is a well-written and informative manuscript.  In the discussion section, I wonder if the authors could further comment on why only 38.5% of screen positive for non-sickling disease infants underwent molecular testing.  Furthermore, could the authors share their thoughts on incorporating molecular testing in newborn screening for hemoglobinopathies.  Other minor comment are below:

1.       Line 26, I suggest replacing “inherited diseases” with “congenital diseases”.

2.       Line 35, I suggest replacing “diagnose” with “identify”.

3.       I suggest using “birth prevalence” instead of “prevalence” when it is calculated based on newborn screening.

4.       Both “abnormal newborn screening results” and “positive newborn screens” are used in the manuscript.  I suggest using “positive newborn screens” consistently throughout the manuscript.

Author Response

Comments:

Response

Line 26, I suggest replacing “inherited diseases” with “congenital diseases”.

Line 26: Edited “inherited” to “congenital disease” as suggested.

Line 35, I suggest replacing “diagnose” with “identify”.

Line 35: Edited “ diagnose” to “identify” as suggested.

I suggest using “birth prevalence” instead of “prevalence” when it is calculated based on newborn screening.

Line 18, 77, 80, 173, 244, 247, 249, 257, 387: “Prevalence” has been edited to “birth prevalence” as suggested.

Both “abnormal newborn screening results” and “positive newborn screens” are used in the manuscript.  I suggest using “positive newborn screens” consistently throughout the manuscript.

Line 143: “Abnormal newborn screening results” have been replaced with “positive newborn screens” to maintain consistency throughout the manuscript as suggested.

Additional changes

Table 1 has been edited for clarity.

Table 2 has been edited and the numbers updated for clarity.

Figure 3 and 4 have also been updated to reflect the overall number of newborn screens performed annually within the state of Iowa.

Figure 5 have been added as a summary flowchart for clarity.

Reviewer 2 Report

Comments and Suggestions for Authors

This study reports trends of rising prevalence of hemoglobinopathies in the newborn screens in a 20-year period in Iowa.  Thalassemias are emphasized, because the authors note that the newborn screening program was designed for sickle cell disease and needs to be adapted to thalassemias. The authors link immigration patterns to these trends in alpha thalassemia, and beta thalassemia. They discuss a standardized algorithm for newborn screening methodologies, diagnosis, and follow up for hemoglobinopathies.

The technical details and the real-world data are interesting and have relevance for other regions of the US. The Discussion section presents comparative merits of technologies and some options for linking patients to care.  The writing is clear and the descriptions in text and tables are detailed. The emphasis is on the laboratory aspects of the newborn screening program but there is also discussion about to improve follow-up. They describe intermittent epidemiologic data collection projects by CDC for SCD health outcomes. They propose similar long-term epidemiologic data collection for thalassemia.

However, the overall process picture is difficult to follow, especially where the ‘missing data’ fit in the process (Tables 1 and 2) and how many babies are lost to follow-up for thalassemia and SCD. Please consider adding some flow-diagrams. Please consider showing in (Figures 3 and 4) the bar-graphs of numbers of SCD and thalassemia detected per year the number of cases missing confirmation.

Minor

Describe the methods used by the short-term follow-up team for retrieving them. Have these methods been adapted for the greater population of immigrant families – e.g. language skills, ethnic community organizations, healthcare champions in the immigrant community?

Lines 193-194  Among the 65 cases of beta thalassemia and its structural variants, only 19 (29%) of these cases were confirmed molecularly.  Please clarify – was the diagnosis sufficiently clear without molecular confirmation?

Author Response

Comments

Response

However, the overall process picture is difficult to follow, especially where the ‘missing data’ fit in the process (Tables 1 and 2) and how many babies are lost to follow-up for thalassemia and SCD. Please consider adding some flow-diagrams. Please consider showing in (Figures 3 and 4) the bar-graphs of numbers of SCD and thalassemia detected per year the number of cases missing confirmation.

Please see the updated Figure 3 and Figure 4 that shows the total number of annual newborn screens in comparison to the annual number of positive SCD (Figure 3) and non-SCD (Figure 4).

Please see the summary flowchart in Figure 5 (newly added).

Describe the methods used by the short-term follow-up team for retrieving them. Have these methods been adapted for the greater population of immigrant families – e.g. language skills, ethnic community organizations, healthcare champions in the immigrant community?

Once our short term follow up team calls out the positive results, the patients are scheduled to a pediatric hematologist for follow up (either at our institution or at Blank Children’s Hospital, Des Moines).  In our institution, we have an in-house genetic counselor that provides genetic counselling to these patients and was able to record these results and follow up molecular testing (if any). These methods have not been adapted specifically for the immigrant community or utilized outside of our institution.

Lines 193-194 : Among the 65 cases of beta thalassemia and its structural variants, only 19 (29%) of these cases were confirmed molecularly.  Please clarify – was the diagnosis sufficiently clear without molecular confirmation?

Line 193-194: Unfortunately, we do not have specific details of the cases that were not seen by the pediatric hematologist at our institution and were not able to verify if the diagnosis was clear and correlated with additional laboratory testing on follow up.

Additional changes

Table 1 has been edited for clarity.

Table 2 has been edited and the numbers updated for clarity.

Figure 3 and 4 have also been updated to reflect the overall number of newborn screens performed annually within the state of Iowa.

Figure 5 have been added as a summary flowchart for clarity.